# Conceptual Framework for Modeling Dynamic Complexities in Produced Water Management

Robert Sabie [1,*], Saeed P. Langarudi [2], Kevin Perez [1], Bruce Thomson [3] and Alexander Fernald [1]

1. New Mexico Water Resources Research Institute, New Mexico State University, Las Cruces, NM 88003, USA; kap470@nmsu.edu (K.P.); afernald@nmsu.edu (A.F.)
2. System Dynamics Group, Department of Geography, University of Bergen, 5007 Bergen, Norway; saeed.langarudi@uib.no
3. Department of Civil Engineering, The University of New Mexico, Albuquerque, NM 87131, USA; bthomson@unm.edu
* Correspondence: rpsabie@nmsu.edu

**Abstract:** This research addresses a gap in the produced water management (PWM) literature by providing a conceptual framework to describe the connections of PWM to regional water budgets. We use southeastern New Mexico as a case study, because the region is facing looming shortfalls in water availability, and oil and gas production generate high volumes of produced water in the region. The framework was developed through expert interviews, analysis of industry data, and information gained at industry meetings; it is supported by detailed descriptions of material flows, information flows, and PWM decisions. Produced water management decisions may be connected to regional water budgets through dynamic complexities; however, modeling efforts exploring PWM often do not capture this complexity. Instead, PWM is most often based on the least expensive management and disposal alternatives, without considering short and long-term impacts to the regional water budget. On the other hand, regional water budgets do not include treated produced water as a potential resource, thus missing opportunities for exploring the impact of potential beneficial reuse. This is particularly important when there is a need to address water shortages in chronically water-short regions of the United States. At the same time, oil and gas production in the western United States is challenged by the need to dispose of large volumes of produced water. The framework is useful for developing improved models of PWM to identify the impact of alternative management decisions on regional water budgets.

**Keywords:** produced water; conceptual framework; modeling; New Mexico

## 1. Introduction

Understanding the effects of produced water management (PWM) options on local and regional water budgets requires novel modeling solutions. Oil and gas (O&G) production provides major economic benefits to oil- and gas-producing states and at the same time generates very large volumes of waste water in the form of produced water (PW)—a mixture of chemical laden flowback water and formation water containing very high concentrations of total dissolved solids. Effective PWM may have positive impacts such as making additional water available for fit-for-purpose use [1] and ameliorates negative impacts such as increased risks to the environment [2]. Current PW management generally consists of disposal in deep saltwater disposal (SWD) wells and reinjection into O&G formations for pressure control and secondary oil recovery. At present high costs, technological limitations associated with PW treatment prohibit significant reuse outside of the oil and gas industry. Increasing reuse of treated PW offers reduced environmental risks to surface and groundwater contamination and may reduce the imbalance between water needs and availability of freshwater supplies. There is a need to methodically characterize the

advantages and disadvantages regarding societal and environmental impacts of difference PWM alternatives.

PWM impacts vary over time and location, thus are hard to track through deterministic modeling. Total PW generated in a region vary over time with nonlinear feedbacks depending on oil and gas production rates, the type of well used to recover the resource, and the source and volume of water used for hydraulic fracturing. PW volumes also vary with well location and depend in part on the depth, length, and geological formations from which a well is producing. PWM decisions regarding disposal also vary by location, because some regions are more adapted to reusing the majority of generated PW in the region for hydraulic fracturing (HF) or secondary recovery of oil (SRO), whereas in other regions, the volume of PW far exceeds the demand for HF and SRO. For example, in the Eagle Ford Shale, Marcellus Shale, and Midland Basin, because PW volumes generated by O&G production are similar to those used for HF or SRO, there is a reduced need for disposal [3]. A contrasting example is the Permian Basin in New Mexico where, in 2021, the industry generated 1543 MMbbl (199,000 acre-ft) of PW while using 28 MMbbl (37,000 acre-ft) of water for HF, only 50 percent of which was PW [4], thus requiring disposal of large volumes of highly saline water in SWD wells. Competition for freshwater supply, increasing drought conditions [5], and tighter regulations limiting PW disposal through injection [6] amplify the need for new sources of water that could potentially be met by treated PW. The first steps have been taken to examine water issues associated with PW in O&G producing regions [7]. Despite most PW in the United States being generated in semiarid Southwest [8], including treated PW in the water portfolio of drought-prone regions is not well understood.

To better understand the dynamic complexities of PW impacts on the water budget, we developed a comprehensive conceptual model framework. Previous research suggested that PWM affects both natural and human processes related to water budgets at local and regional levels through multidimensional feedback systems [9]. Theoretically, decisions made at the individual (micro) and system (macro) levels have complex interactions; however, there are few examples in the literature where these interactions are explicitly identified. Several studies considered the major components of the PWM system to provide guidance on some of the specific components. Beattie, et al. [10] created a model that optimizes PWM with a focus on infrastructure. Geza, et al. [11] and Ma, et al. [12] described a decision–support tool for optimally selecting a treatment technology based on PW quality, the intended beneficial use, and the underlying economics. Sullivan Graham, et al. [13] discussed a generalized decision process for identifying PW recycling opportunities and the importance of considering PW supply as part of a water budget. Thomson and Chermak [4] compiled detailed information on trends of PW generation and management in New Mexico and described the economic considerations for PW reuse. Baca, et al. [14] analyzed the state and federal legal and regulatory framework for PWM and provide guidance on navigating the permitting process for PWM. Scanlon, et al. [3] quantified PW volumes and quality for the major United States oil-producing regions and outlined the potential beneficial reuse for irrigation, municipal use, surface water discharge, groundwater recharge, hydraulic fracturing, and industrial use. A report by the Groundwater Protection Council [15] discussed the important considerations for reusing PW including modeling of the health and environmental risks associated with leaks and spills, and it suggested a gap in modeled implications of beneficial PW reuse on energy demand. Langarudi, et al. [9] suggested developing and using hybrid modeling approaches that simultaneously capture micro- and macro-level dynamics of PWM to track the impact of alternative strategies on the water budget. Recently, Tidwell, et al. [16] published a report that described the development of a proof-of-concept system dynamics model to assess trade-offs between economic, social, and environmental outcomes of PWM. Despite all these exploratory efforts, there is a still a lack of a comprehensive, generic conceptual framework for simulation modeling of how PW management and reuse interactions impact regional and local water budgets.

This paper contributes to the PWM literature by developing a conceptual framework that forms a basis for simulation modeling of dynamically complex problems of PWM. Our framework was developed to answer the question: "How does PWM connect to a regional water budget?" Ultimately, PWM decisions aim to balance the economic drivers with the goals of reducing water scarcity, avoiding increased seismicity, and minimizing the risks of contamination and exposure. Before PWM can be effectively modeled, there must be a comprehensive understanding of its interactions and complexities. This paper constitutes a major step toward in filling this gap by providing guidance for future computational modeling of these PWM complexities. Section 2 describes our research methods and explains our synthetic conceptual framework. In Section 3, we discuss how this conceptual framework can be used to help PWM modeling. Section 4 provides a summary and conclusions.

## 2. Methods

Eddy and Lea counties constitute the major O&G producing regions of the Permian Basin in southeastern New Mexico, USA, and were used as a case study. We developed our conceptual framework by synthesizing the knowledge we gathered from previous studies and interviews conducted with industry and regulatory experts.

### 2.1. Case Study Area

The location of Eddy and Lea counties is shown in Figure 1. These two counties have been the focus of several recent studies [9,17–19] because they are two of the largest oil- and gas-producing counties in the United States and because of the continued stress on surface and groundwater resources. The region is arid and receives less than 40.6 cm (16 in) of annual average precipitation. The counties overlie the western extent of the Permian Basin, which in 2017 was estimated to have the world's largest unconventional oil play [7]: in 2019, the region had approximately 21,907 oil wells and 2850 gas wells [18]. Within the study region, estimates of PW reuse within the industry for HF and SRO range from 50 percent [4] to 58 percent [18], the remaining PW being disposed of in SWD wells. The region has experienced an increase in seismicity likely connected to the disposal of PW in SWD wells [20]. New regulations from the New Mexico Oil Conservation Division (NM OCD) are expected to place limitations on the volumes and pressures allowed for PW injection in SWD wells. The median distance from an oil and gas well to a SWD well is approximately 1.1 miles [21]. The area has limited surface water, and groundwater supplies of the High Plains Aquifer are showing signs of rapid depletion [22]. The regional water budget is directly impacted by the oil and gas industry because large volumes of freshwater (water with total dissolved solids (TDS) of <1000 mg/L) and brackish water (TDS between 1000 and 10,000 mg/L) are used for HF although recent data reported to the NM OCD show a marked substitution of PW for freshwater for HF in NM [4]. We aim to show with the conceptual model that an integrated perspective is required to determine the positive or negative impacts on the local and regional water budget as a result of various PWM decisions.

We interviewed PWM experts from academia and industry to gain more insight into the structure of the produced water management system in New Mexico. Our application to conduct interviews was approved by the New Mexico State University Institutional Review Board (IRB). We conducted eight semi-structured 90-minute interviews through video conference. The questions (see Appendix A) were designed to determine how PWM decisions are made, the time scale in which management decisions occur, the risks of mismanagement, and the foreseen opportunities for better management. Eight participants were interviewed, with expertise in the areas of water treatment, petroleum engineering, water law, economics, environmental engineering, and civil engineering.

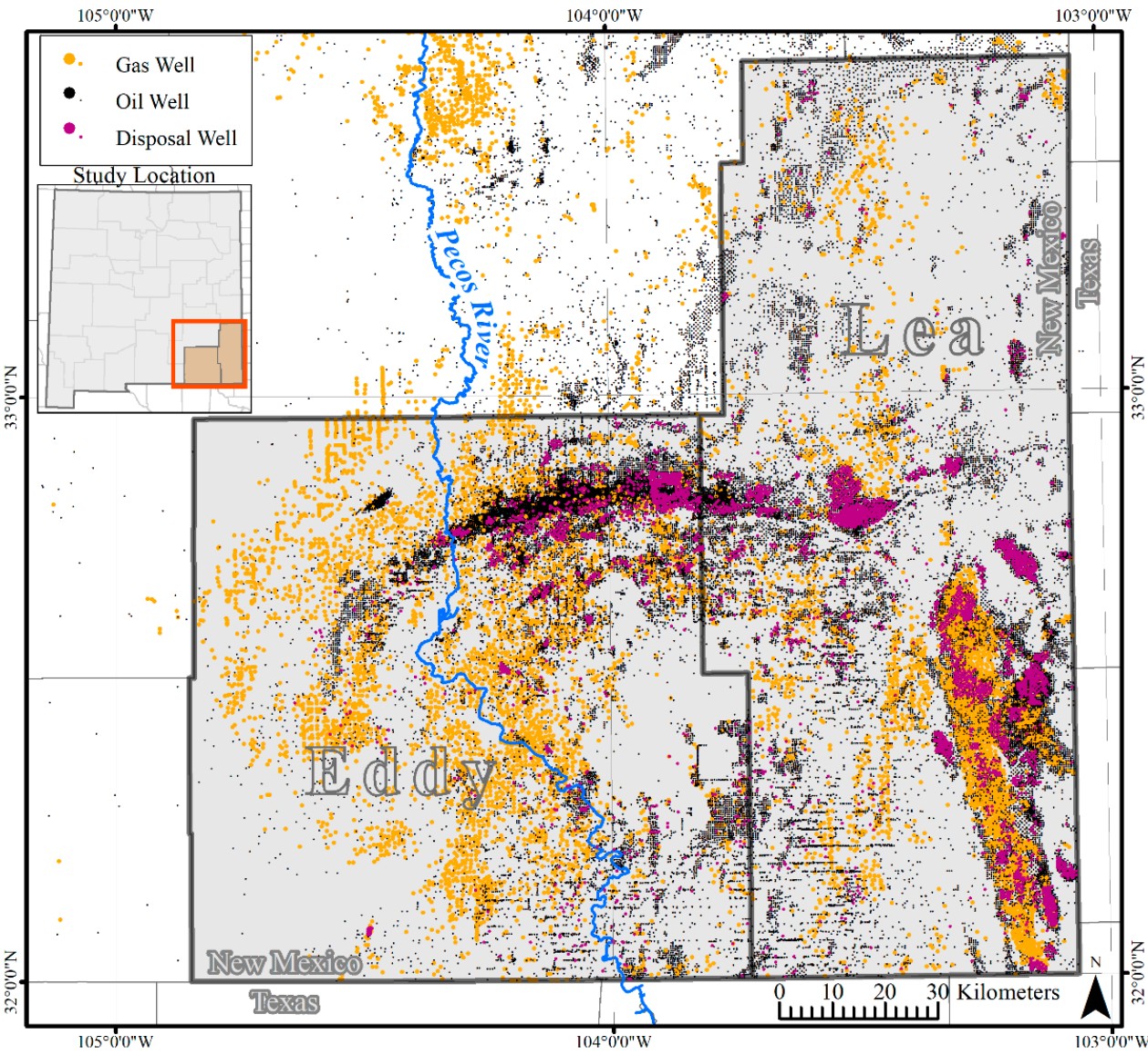

**Figure 1.** Oil and gas wells along with disposal wells in Eddy and Lea counties, New Mexico.

### 2.2. Synthetic Conceptual Model

We developed a conceptual framework (Figure 2) useful for constructing a potential dynamic simulation model to account for key complexities associated with PWM and its relationship to local or regional water budgets. This framework can be applied to any case of PWM impact on a water budget regardless of its local singularities. Each of the linkages presented within the framework is generic. In Figure 2, the solid arrows represent material flows (freshwater, produced water, residual solids, etc.) while the dashed arrows represent information flows (cost, regulations, etc.). Trapezoids represent information variables, whereas hexagons represent material variables. Variables with a dark background are exogenous (independent of any other model variables). Management decision options are in rectangular boxes. The following subsections describe the major interactions of Figure 2 and explain the relationships to the regional water budget (defined below).

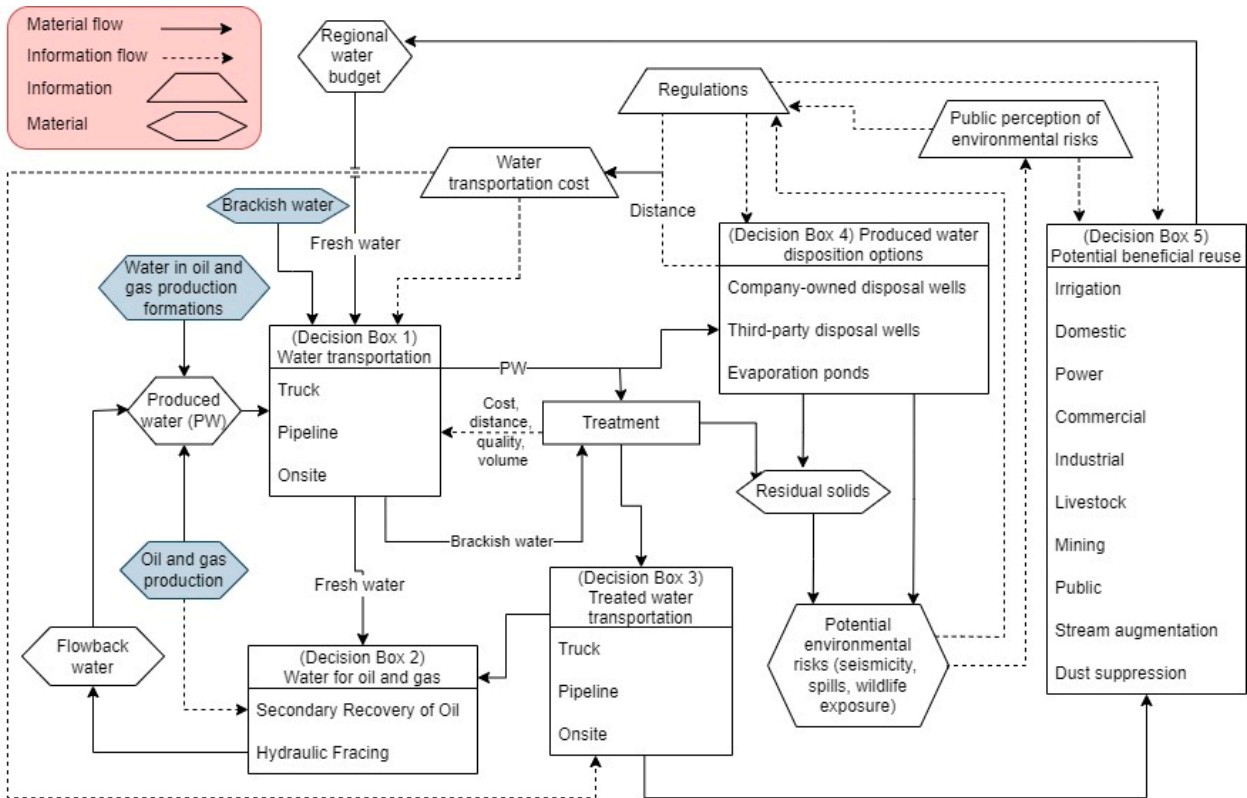

**Figure 2.** Conceptual framework for simulation modeling of PWM impacts on water budget.

### 2.2.1. Regional Water Budget—Material Flow

This study identifies and describes the relationships between PWM and a regional water budget, including both fresh and saline water resources. A regional water budget constitutes a mass balance that accounts for all the inflows and outflows (Equation (1)) of water within a defined region and over a defined period of time. The variables included can be aggregated to the major components (Equation (2)) or disaggregated to include specific withdrawals (Equation (3)). We adopted Equation (3) from Peterson, et al.'s New Mexico Dynamic Statewide Water Budget (NMDSWB) model [23] and included the term $PW_t$ to account for the addition of treated PW to the water budget. NMDSWB offers a good example of a disaggregated water budget with detailed water accounting and provides a comprehensive analysis of fresh surface and ground water resources at the state-wide, regional, and county levels that could be readily extended to include saline and PW resources. Note that only after PW is treated to a sufficient quality can it be included as part of a water budget.

$$\text{Change in storage} = \text{inflows} - \text{outflows} + \text{source} - \text{sinks} \tag{1}$$

$$\Delta S = Q_{in} - Q_{out} + P - ET \tag{2}$$

P = precipitation
$Q_{in}$ = water flow into the watershed
ET = evapotranspiration
$\Delta S$ = change in water storage
$Q_{out}$ = water flow out of the watershed

$$\Delta HSDS - \Delta GW_s = SW_d + P_r + SW_r + PW_t - GW_i - ET_h \tag{3}$$

$\Delta HSDS$ = change in Human Storage and Distribution System over a given period of time
$\Delta GW_s$ = change in groundwater storage

$SW_d$ = surface water diversions to human use
$P_r$ = precipitation directly into reservoirs
$SW_r$ = surface water returns from human use
$PW_t$ = treated produced water
$GW_i$ = groundwater infiltration to groundwater from reservoirs and streams
$ET_h$ = reservoir evaporation

In our conceptualization, the contribution of PW to the water budget occurs through two main connections. The first connection is through the water used for oil and gas production to facilitate HF and SRO. Water is transported (Box 1) and used in oil and gas production (Box 2). If the water used is freshwater, then this is a withdrawal from the resource. If PW or brackish water is used instead it reduces the freshwater requirements for that purpose. The second connection to the regional water budget is when PW is treated and transported (Box 3) for a beneficial use (Box 5), thus adding water to the fresh water budget. For context in the same study area, Jiang, et al. [18] estimated $85.9 \times 10^6$ m$^3$ of PW was being disposed in SWD wells in 2019, so that approximately $113.3 \times 10^6$ m$^3$ of PW was being reused, virtually all of which was for SRO. Thomson and Chermak [4] show that in recent years the fraction of PW used for SRO is decreasing due to the inability to circulate water through the very tight formations being developed by unconventional oil recovery technologies. This trend will require development of additional future SWD wells. Sabie, et al. [21] estimated that $45.5 \times 10^6$ m$^3$ of PW was potentially available for beneficial reuse outside the oil and gas industry within the study area. There are limited data on water sourcing for oil and gas production in New Mexico; however, data generated in the last two years shows an increasing trend towards the substitution of high-salinity PW for fresh and brackish water for HF operations [4]. Currently, brackish water and PW are not considered as part of the country or regional water budgets because the New Mexico Office of the State Engineer does not associate a water right with either of these water types.

### 2.2.2. Produced and Flowback Water—Material Flow

The volume of produced water generated depends on four main factors: oil and gas production rates; volume of water from the O&G formations; whether the O&G is recovered by conventional or unconventional methods (i.e., vertical wells versus horizontal wells); and the volume of water used for HF and the fraction returned as flowback. O&G production is the primary driver of PW generation but how much is generated alongside oil and gas production depends on the flowback volume and the water-to-oil ratio associated with the geologic formation. Flowback water is a combination of geologic formation water and the water injected during the HF process. They are usually managed together and reported as PW (Equation (4)). Within this definition, when freshwater or brackish water from the regional water resource is used for oil and gas operations, it becomes PW as flowback.

$$PW = \text{geologic formation water} + (\text{freshwater} + \text{brackish water} + PW) \qquad (4)$$

Secondary recovery of oil (Box 2) is a practice of using water, gas, and sometimes other fluids such as steam or solvents to displace residual oil after initial recovery to increase production. PW used for this purpose stays outside of the regional water budget.

### 2.2.3. Water Transportation—Decision

Water (whether PW, freshwater, or brackish water) is transported (Boxes 1 and 3) either by trucking or through pipelines. It may also be generated or disposed of onsite. Onsite is an option because some O&G operations have their own onsite freshwater wells, SWD wells, and/or water treatment facilities. Trucking is assumed to have the highest costs and varies with the cost of fuel and distance driven on a road network. Pipelines have a large initial cost but are assumed to have a lower transportation cost once constructed.

Transportation costs are an important consideration affecting the feasibility of beneficial reuse of PW.

Decisions regarding PW transportation primarily depend on the costs of transportation along with a potential demand for reuse and the cost of disposal. That is, it should either be disposed (Box 4), or treated and reused (Boxes 2 and 6). Regardless of how it is managed, PW must first be transported (Box 1) either to a treatment plant or a disposal point determined through a cost-benefit analysis. Box 1 also defines how fresh and brackish waters are transported. The decisions made in Box 1 are based on the cost, distance, quality, and demand for different types of water. Once produced or brackish waters are treated, similar transportation decisions define how the treated water is delivered to its intended destination (Box 3). Note that transportation costs depend on several factors including the distance between the water's origin and its point of use as well, topography (large elevation differences may make pumping infeasible), access to right-of-way for piping systems, and environmental regulations. In some cases, producers might have contractual relationships with mid-stream companies that make it more cost-effective to transport PW longer distances. Regulations can affect the cost of transportation. For example, if the perceived risk of PW spills from trucks increases, stricter regulations might be imposed on the road transportation of PW, implying a higher cost for such transportation methods.

Figure 3 identifies factors affecting the water transportation costs. Manhattan distances (the sum of absolute difference between the measures in all dimensions of two points) between origin and destination of PW are the main factors driving these costs. When different transportation methods are compared, other factors come into play; for instance, the relative cost of using pipelines also depends on the pipeline development costs (obtaining right-of-way, layflat vs. hard pipe, etc.), availability of each transportation means, and environmental regulations.

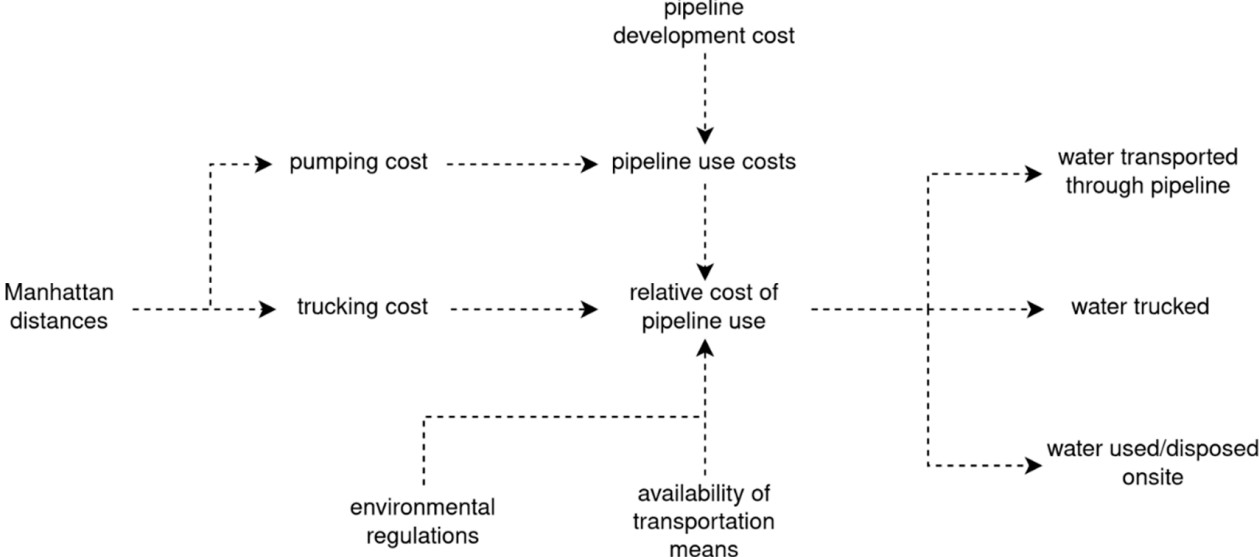

**Figure 3.** Water transportation decision tree.

2.2.4. Produced Water Disposal—Decision

The decisions for what and where to dispose of PW depend on several factors. The first decision is typically based on a benefit-cost assessment: oil companies could dispose of the PW (Box 4) by injecting it in their own SWD wells or in those owned by third parties. Another option is to dispose of PW in evaporation ponds. All the options impose environmental risks such as spills. Injections into SWD wells increase the risk of seismicity [24], while disposal in evaporation ponds will generate residuals that could be hazardous to the ecosystem and wildlife [25]. These decisions will determine the Manhattan distances, which affects transportation costs as mentioned earlier.

The other factor that affects transportation costs is environmental regulations, which depend on the current and past states of the environmental risks imposed by seismicity, PW spills, and wildlife exposure.

Produced water treatment generates solid residuals, which require handling and disposal and may pose an environmental risk. A significant portion of the treated produced water remains and will be reused within the oil and gas industry (Box 2): the rest will be reused by other potential users (Box 5).

Ultimately, PWM affects the water budget by increasing the supply options listed in Box 5. Depending on the level of treatment provided, PW reuse (Box 5) might include all water use categories (i.e., irrigation, domestic, power, commercial, industrial, livestock, mining, and public) as well as stream augmentation and dust suppression. Potential PW reuse depends on the quality of the treated water that is subject to state and/or federal regulations. High-quality water is required for domestic, irrigation, livestock, public, stream augmentation, commercial, industrial, mining, and power, whereas water used for dust suppression typically may not need to meet stringent criteria. Other potential beneficial reuse options exist, but dust suppression and stream augmentation are two options frequently mentioned by industry and regulators in the study region. The quality of water required for each use category is affected by the available and accessible treatment technologies.

### 2.2.5. Treatment—Decision

In our framework, treatment occurs within a decision box, although we do not list the numerous treatment choices: three of our interview participants suggested that over 70 options may be available for PW treatment. The decisions taken for selecting an appropriate treatment technology have been the subject of previous research [11,12]. In general, the treatment decision will be based on the cost and performance of treatment technologies appropriate to the quality of PW including energy costs, the expected use of the treated water, the value of the treated water including the customers' willingness to pay for that water, management and disposal options for residuals from the treatment process, and the expected operating life of the treatment facility.

### 2.2.6. Regulations—Information Flow

Regulations are an important information flow and the placement within our framework is useful for evaluating the effect of different PWM policies and regulations on management strategies and regional and local water budgets. In our conceptualization, regulations are a function of both real environmental risks and the public's perceived environmental risks. Regulations also have an influence on the transportation method (Boxes 1 and 3), the PW disposal decisions (Box 4), and on beneficial reuse options (Box 5). Regulations on pipeline permits impact the construction of new pipelines. Potential environmental risk may lead to new regulations.

The immediate environmental risks associated with PW focus on spills and wildlife exposure. According to a report from the US Environmental Protection Agency a majority of the spills related to hydraulic fracturing are small spills of PW (<1000 gallons) from storage facilities and are caused by human error [26]. Many spills on the surface occur from leaks in pipe networks or bad connections when filling tanker trucks. Evaporation ponds are a risk to birds that are attracted to the water bodies [27]. Habitat loss and fragmentation, introduction of invasive species due to disturbances, and changes in the variety and abundance of various species are other environmental risks [28]; in some cases they can trigger additional regulations such as the Endangered Species Act.

Although many regulations are in place to protect against environmental and human exposure, the self-reporting of spills by industry raises public skepticism on the accuracy of the reporting. The New Mexico regulatory agencies seldom perform field inspections of leaks and spills due to staffing shortages. One of the experts participating in this study stated that the biggest challenge in reusing treated PW in New Mexico is addressing the public's concerns over human and environmental exposure. We attended several industry

meetings where the public perception of risk was discussed as the biggest challenge in beneficial reuse of treated PW. In New Mexico, regulations currently do not allow for reuse of treated PW outside of the oil and gas industry, although new regulations are under development to permit this in the future.

The choice of disposal option is impacted by regulations leading to implications for the regional water budget. For example, both New Mexico and Texas recently implemented more stringent reporting requirements for SWD wells to reduce risks of induced seismicity caused by deep well injection of PW. These regulations are having an immediate impact on PW disposal, where in December 2021 SWD well permits were suspended in Gardendale Seismic Response Area in northwest Midland County, Texas, and in May 2022 a large producer was fined USD 2.2 million for non-compliance with disposal regulations in New Mexico. As the cost of disposal in SWD wells increases due in part to the regulations, increased demand on existing SWDs, and increasing volumes of PW, the cost of treating PW for beneficial reuse is becoming a viable economic alternative to disposal.

2.2.7. Potential Beneficial Reuse—Material Flow

There are numerous potential benefits from reuse of treated PW. The United States Environmental Protection Agency has exemptions to the National Pollutant Discharge Elimination System, and states in the National Water Reuse Action Plan (WRAP) that "West of the 98th meridian, produced water that is of good enough quality, and that has a use in agriculture and wildlife propagation and is put to such use during periods of discharge, can be discharged for beneficial reuse." In New Mexico, neither raw nor treated PW are considered as part of local or regional freshwater budgets. Therefore, beneficial reuse could positively impact the regional water budget by providing an additional source of water for the region. Because O&G operations are transient, reuse of treated PW will not be associated with a new water right.

Reuse of PW within the oil and gas industry is currently allowed and encouraged in New Mexico. Jiang, et al. [18] estimated in 2019 that the oil and gas industry currently reuses approximately 58 percent of PW for either HF or SRO.

Although water quality standards for beneficial reuse of treated PW in New Mexico are not yet established, the New Mexico Produced Water Research Consortium is currently investigating treatment methods to assist in the development of regulations to permit reuse for purposes outside of the industry up to and including augmenting public water supplies.

When water quality standards are established in New Mexico, treated PW could potentially be a water source for the uses listed in Box 5. Each of the potential beneficial use cases will require a separate analysis for determining specific water quality requirements, the volume of water demand, the economic feasibility, and the spatiotemporal supply–demand situation. We suggest any modeling efforts of PWM consider not only the economics to the oil and gas industry but also the broader implications to volume of water within the regional water budget, the forgone opportunities of freshwater use compared to treated PW use, and the long-term sustainability of the water budget.

**3. Discussion and Future Work**

The conceptual framework presented in this paper is a useful start to understanding the linkage between PWM and regional water budgets; however, considerable work remains. The exogenous components of the framework are not described in detail in this conceptualization but are nonetheless critical to the development of a functional model. Particularly, the price of O&G and the cost of its production ultimately drives the industry's activities and thus driving the whole framework. Despite several decades of O&G production in the Permian Basin [3], inclusion of treated PW into a regional water budget should be done with caution as the availability is driven by many factors and it is not a long-term water source. It cannot be overstated that treatment costs of PW are not trivial—often ranging between USD 2.92/m$^3$ and USD 5.45/m$^3$ for PW with a salt concentration of 90 g/L from Osipi, et al. [29] and Xu, et al. [30]. In order for beneficial reuse of treated PW

to occur, the cost of treatment needs to be similar to the cost of disposal and the cost of acquiring and transporting fresh water to the point of use.

Constructing a computational model of the framework presented here faces more challenges than traditional system dynamics models or agent-based models. On one hand, the model requires to define objects with more attributes than the classical stock and flow structures to perform the water mass balance calculations; for example, the decisions regarding water transportation depend on both physical factors such as the distance between its source and fate, but also on institutional and economic considerations such as regulatory constraints on disposal or reuse, ownership and availability of SWD well capacity, and competition among mid-stream companies that provide PW transportation, treatment, reuse, and disposal services. A hybrid system dynamics (SD) and agent-based modeling (ABM) approach, as suggested by Langarudi et al. [9], can help to represent the complex spatial and temporal dynamics of the framework by driving some of the SD variables by ABM interactions and using some of the SD model outputs as variables in the ABM. However, a hybrid model makes the traditional software for dynamic simulation insufficient. On the other hand, a system dynamics model must include enough information to set the initial values and parameterize the input and output relationships.

Although the model framework is assumed to operate over discrete timesteps, the specific duration is not defined, although this would be important for some of the potential treatment and reuse scenarios. For example, modeling reuse of treated PW for agriculture, by far the largest consumptive user of water in southeastern New Mexico, would require consideration of the intra-annual variation in crop water demand [21]. Planning decisions for PWM, e.g., building pipelines and treatment infrastructure, often occur over a longer time span. Constructing large infrastructure projects such as pipelines and treatment plants usually requires many years from conception to completion. Another temporal element is the temporary storage that occurs between treatment and reuse, which is the case for both reuse within and outside of industry.

Future improvements to this conceptual framework could be made with the inclusion of a life cycle assessment (LCA) of the different PWM decisions and the inclusion of additional site-specific variables. LCA could be used to provide information into a dynamic model or could be used in parallel for assessing the environmental impacts of each PWM decision. The conceptual framework presented in this study is general; however, there are other potential geographically specific variables that could strengthen modeling efforts. One specific example is the inclusion of mineral recovery from the PW treatment concentrates, where it has been researched for acid mine drainage [31] and brine management [32] as a variable for potentially offsetting the cost of treatment.

The impacts of potential treated PW reuse to the regional water budget likely depends on the beneficial reuse. Agriculture is the largest water user in the study region, but the concerns over water quality, the level of treatment required, and the proximity of irrigated lands to the supply of treated PW are barriers to adopting treated PW as a water source. The agricultural industry is also the least able to afford the high costs of treating and transporting PW. This point emphasizes the importance of the public perception of risk, treatment options, and distance within our framework. Another example of potential treated PW reuse is augmenting the Pecos River: New Mexico is required to deliver water to Texas via the Pecos River as part of the Pecos River Compact and, in some years, pays to pump groundwater to meet the delivery requirement. Depending on PW transportation and treatment costs, it may be feasible to use treated PW to increase the delivery of fresh water to Texas, a management strategy that can be evaluated by this model.

## 4. Conclusions

We constructed a conceptual framework of a model to connect PWM decisions to a regional water budget using southeastern New Mexico as a case study. The framework was developed through numerous iterations and conversations with academic and industry experts. To the authors' knowledge, this is the first explicit conceptualization directly

connecting PWM decisions to a regional water budget. As water shortage persists and options for PW disposal continue to be limited, it is important for future models to be developed based on this framework in order to capture the broader impacts of PWM decisions, and particularly, beneficial reuse on regional water budgets.

**Author Contributions:** Conceptualization, R.S., S.P.L., K.P., A.F. and B.T.; investigation, R.S. and S.P.L.; data curation, R.S., S.P.L. and K.P.; writing—original draft preparation, R.S. and S.P.L.; writing—review and editing R.S., S.P.L., A.F. and B.T.; visualization, R.S. and S.P.L.; supervision, S.P.L. and A.F. All authors have read and agreed to the published version of the manuscript.

**Funding:** This research was funded in part by the New Mexico Universities Produced Water Synthesis Project and by the U.S. Department of the Interior, Geological Survey 104b program award number G21AP10635-00, through the New Mexico Water Resources Research Institute.

**Institutional Review Board Statement:** The study was conducted according to the guidelines of the Declaration of Helsinki, and approved by the Institutional Review Board of New Mexico State University using protocol 45 CFR 46.110(b)(2) and was approved 27 July 2021.

**Informed Consent Statement:** Informed consent was obtained from all subjects involved in the study.

**Data Availability Statement:** All processed data used in the study have been shown in the article. Raw data may be available on request from the corresponding author.

**Acknowledgments:** We thank the editors and reviewers for their valuable comments and suggestions to improve the quality of the paper.

**Conflicts of Interest:** The authors declare no conflict of interest.

## Appendix A

Interview Questions:

What is your role in produced water management?

In your opinion, what are the biggest challenges in produced water management?

How do you think these challenges evolve over time (will they change for better or worse)?

Please describe your opinion of what the best options are for managing produced water.

In your opinion, what are the biggest opportunities for produced water management?

How do you think these opportunities evolve over time (will they change for better or worse)?

What percentage of the water used for hydraulic fracking comes from agriculture (do you have an estimated volume per well?)?

Could you describe some of the routine (daily, monthly, seasonal, annual) decision-making processes in produced water management?

What type of information/input (do you use optimization models; do you use consultants) do you use for decision making?

What is your perceived accuracy of the data used in making decisions on produced water management?

How does public perception influence produced water management decisions?

How and why has perception changed over time?

What are the risks associated with sharing data?

What are your perceived risks of mismanaging produced water?

Please describe characteristics of successful produced water management.

Please describe characteristics of failures in produced water management.

If there were two questions you would like answered about produced water, what would they be?

Is there anyone you think we should interview to get more information?

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
