# Peer review of "Conceptual Framework for Modeling Dynamic Complexities in Produced Water Management"

_water, doi:10.3390/w14152341_

Round 1
Author Response
R1: Modeling dynamic complexities in produced water management was carried out.
It is an interesting study and contributes to the development of the produced water management (PWM). I therefore suggest acceptance of this article before several minor revisions are made.
Response: The authors would like to thank the reviewer for reviewing and providing feedback to improve our manuscript. Below are detailed responses to each of the suggestions.
R1: Produced water management decisions may be connected to regional water budgets through dynamic complexities, however, modeling efforts 17 exploring PWM often do not capture this complexity. However, southeastern New Mexico as a case study was investigated in this study, the author should compared this work with other analogous works linked with other cities.
Response: Although the case study focuses on southeastern New Mexico, the framework was developed to be adaptable to other regions with PWM issues. We agree there may be other similar scenarios where treated wastewater is reused in cities, but the drivers within those systems would likely be much different, thus outside the scope of this study.
R1: The framework was developed through expert interviews, analysis of industry data, 14 and information gained at industry meetings and is supported by detailed descriptions of material 15 flows, information flows, and PWM decisions. How about other factors to support the framework?
Response: The factors we used in our conceptual framework are generalizable to most cases of PWM and water budgets and were derived from industry reports, scientific literature, and expert interviews. We suggest these are the main factors. We do agree there could be additional factors that could support the framework, however, they would likely be geographically specific. To address this comment, we added a short paragraph in the discussion section describing how mineral recovery from the solid waste stream could help offset the treatment cost.
Reviewer 2 Report
The paper discussed about the PWM framework for the New Mexico region. The authors addressed the area in specific and proposed a detailed framework for the PWM. The paper shows good outcome, however, minor modifications are needed prior to publication.
1. The developed conceptual framework are detailed and well constructed. However, the author should justify the reasons behind the framework, considering the needs of each stakeholder. And generally, there are meaning of each shape which the author should get the right definition.
2. Some of the equations stated are not cited from any reference. Please add.
3. Author also mentioned about the modeling dynamic that will be implemented. However, I cannot see the content in the paper. Please further elaborate or remove the word if it is not linked.
Author Response
R2: The paper discussed about the PWM framework for the New Mexico region. The authors addressed the area in specific and proposed a detailed framework for the PWM. The paper shows good outcome, however, minor modifications are needed prior to publication.
Response: The authors would like to thank the reviewer for reviewing and providing feedback to improve our manuscript. Below are detailed responses to each of the suggestions.
R2: 1. The developed conceptual framework are detailed and well constructed. However, the author should justify the reasons behind the framework, considering the needs of each stakeholder. And generally, there are meaning of each shape which the author should get the right definition.
Response: The authors would like to better respond to this comment, however, are unclear what the reviewer is asking. The reason behind the framework is to conceptualize the connection of PWM to regional water budgets. The shapes for the diagram are described in section 2.2 and also included in the legend.
R2: 2. Some of the equations stated are not cited from any reference. Please add.
Response: Equations 1 and 2 are generally accepted equations for water budgets and we feel no citations are necessary. We added a citation to Equations 3 as it was adopted from Peterson et al 2019. Equation 4 is introduced in this paper by the authors.
R2: 3. Author also mentioned about the modeling dynamic that will be implemented. However, I cannot see the content in the paper. Please further elaborate or remove the word if it is not linked.
Response: The authors are unclear on which word the reviewer is suggesting to elaborate on or remove. The framework we developed provides the background for building a dynamic model of PWM impacts on a regional water budget. The modeling framework is designed to model “dynamic” complexities because we are suggesting there would be non-linear changes in the components of the PWM system over time.
Reviewer 3 Report
This manuscript presents very useful produced water management conceptual model. Indeed, the reuse of PW is an important topic for O&G and as the sustainability aspects for the upstream is a topic of a great importance.
Some minor comments can be found below:
1) Can you suggest which PW treatment technology can meet the required cost efficiency? Membranes?
2) Would you consider to perform the Life Cycle Assesment for some PWM solutions in future?
3) Are there any possible upsides for produced water treatment? E.g. if they contain Lithium ions and thanks to very rapidly growth of Lithium market, the PWM business model can have some additional benefits?
Author Response
R3: This manuscript presents very useful produced water management conceptual model. Indeed, the reuse of PW is an important topic for O&G and as the sustainability aspects for the upstream is a topic of a great importance.
Response: The authors would like to thank the reviewer for reviewing and providing feedback to improve our manuscript. Below are detailed responses to each of the suggestions.
R3: Some minor comments can be found below:
R3: 1) Can you suggest which PW treatment technology can meet the required cost efficiency? Membranes?
Response: The authors are familiar with research that matches the intended PW reuse with the available treatment technologies based on the feedwater quality. We cited Geza et al. 2018 and Ma et al. 2018 as sources for this type of analysis. Because our goal was a generic framework, singling out any one particular technology would limit the flexibility of the modeling framework. These pieces of information are captured as an information flow from the treatment decision box in our framework diagram.
R3: 2) Would you consider to perform the Life Cycle Assesment for some PWM solutions in future?
Response: We think this is an interesting idea as either a parallel analysis, or something that could feed information into a dynamic model. We have added a few sentences to consider LCA as a future addition to the PWM conceptualization.
R3: 3) Are there any possible upsides for produced water treatment? E.g. if they contain Lithium ions and thanks to very rapidly growth of Lithium market, the PWM business model can have some additional benefits?
Response: We agree that capturing some of the minerals from the solid waste stream could offset the cost of treatment. We added short paragraph in the discussion section describing this as a potential addition to the framework when PW contains economically extractable minerals.